# A Double Hurdle Estimation of Sales Decisions by Smallholder Beef Cattle Farmers in Eswatini

**Sicelo Ignatius Dlamini** [1] **and Wen-Chi Huang** [2,*]

[1] Department of Tropical Agriculture and International Cooperation, National Pingtung University of Science and Technology, Pingtung 912, Taiwan; frsicelodlamini2@gmail.com

[2] Department of Agribusiness Management, National Pingtung University of Science and Technology, Pingtung 912, Taiwan

* Correspondence: wenchi@mail.npust.edu.tw; Tel.: +886-8-7703203 (ext. 7824); Fax: +886-8-7740190

**Abstract:** Beef cattle farmers are in an ideal position to advance their income through marketing; however, the subsector is characterized by low market participation. Wealth preservation and prestige from cattle accumulation outweigh market incentives, thereby jeopardizing the integration of farmers into organized market systems. Therefore, the study was set to examine the determinants of farmers' sales decisions in cattle marketing. Understanding determinants of sales decisions is an indispensable base for establishing sustainable development policy frameworks that maximize rural economic growth. Descriptive statistics and a double-hurdle model were applied on cross-sectional data collected from 397 farmers through personal interviews aided by a structured questionnaire. Herd size (74.1%), ecological zone (32.4%), slaughters (22.1%), pregnant cows (18.2%), experience (15.0%) and breed type (11.4%) revealed statistically significant effects on the probability of market participation. The key determinants of the level of market participation ($p < 0.01$) included extension, married marital status, pasture availability, cows, heifers, market distance, market information and market channel 2 (individual). Education, experience, non-farm income, expenses and laborers were significant at $p < 0.05$. Widowed marital status and market channel 1 (processor) were found to be significant at $p < 0.1$. Extension adjustments and institutionalization of market linkages are recommended to assist farmers in increasing marketable surplus.

**Keywords:** beef cattle; market participation; level of market participation; double hurdle

## 1. Introduction

Since meat developed into a valuable commodity, the demand for livestock food products has increased as people shift from plant to animal-based protein sources. Such demand and preference shifts are mainly driven by the interplay between population growth and increase in income [1,2]. The projected 2.3 billion additional world population growth by 2050 demands that meat production increases by 200 million tons to 470 million tons by 2050 [3].

Considering that about 72% of the projected increase in meat production will be consumed in developing countries, the demand increase and preference shifts present improved market opportunities for livestock farmers in developing countries [4,5]. Among the types of meat, beef is an economically important commodity that commands a high market price per unit. Thus, beef cattle farmers are at an ideal position to improve their livelihoods through active participation in the radically growing meat industry [6].

Recent beef consumption statistics in Eswatini indicate that total beef consumption in 2017 amounted to 11,256.36 tons, against the domestic production of 8210.04 tons [7]. This reflects a domestic shortfall of 3046.32 tons at a value of 59,403,240 Emalangeni (Eswatini currency denoted by E) based

on the lowest local beef carcass grade price of E19.50/kg. In addition, exports of prime beef cuts to the European Union market declined from 703.25 tons to 310 tons (−55.92%), creating a further export business opportunity for smallholder farmers.

Taking advantage of the domestic and export agribusiness opportunities requires radical transformation in market participation by smallholder farmers [8]. However, cattle marketing in Eswatini is characterized by low market participation through distress sales. The notions of wealth preservation and prestige from cattle accumulation and ownership outweigh the promise of market incentives and income generation. This has jeopardized the integration of smallholder farmers into organized market systems, thus arresting the potential of markets to contribute meaningfully towards the amelioration of rural livelihoods. Moreover, since cattle marketing is practicalized through offtake, low market participation is central to the sustainability discussion in livestock production.

Our consideration of market participation in the context of sustainability is twofold— economically and environmentally. Economically, low market participation contributes to livestock loss of market value since cattle are often sold at old age. This undermines the economic sustainability of the livestock enterprise due to eroded market incentives and reduced income. Environmentally, low market participation promotes overstocking and overgrazing, which exacerbate pasture depletion and land degradation. Studies have also indicated that land degradation, in Eswatini, is more severe on grazing lands than on crop fields [9,10]. In this regard, market participation is a tool for sustainable livestock production systems.

The ultimate climax of vibrant market participation culminates with a significant contribution to the international development priority agenda of food security and poverty alleviation [11]. Since rural livelihood is largely dependent on subsistence agriculture, farmers' market participation is a crucial component for rural development [12]. In cattle farming, active market participation has been identified as a potential vehicle for creating comparative advantages that increase productivity and produce quality [13], which in turn promote specialization and market-orientedness. This makes smallholder farmers, the majority of whom live under absolute poverty [14], to be the prime beneficiaries of the effects of market participation. Hence, advancing market participation is a salient mechanism for sustainable food security, poverty reduction and economic growth in developing economies [15,16].

Furthermore, understanding the factors influencing farmers' sales decisions not only ensures functional food value chains, but is a cornerstone for policy and developmental programs aimed at maximizing welfare gains for the poor. It provides critical information for establishing market linkages for the incorporation of smallholder farmers into market systems to stimulate sustainable domestic food production [17]. This is integral for sustainable food security and the improvement of livelihoods [18].

Considering that 63% of the population in Eswatini live in poverty [19], most of whom are smallholder beef cattle farmers, advancing vigorous market participation in cattle marketing fits well in the sustainable rural livelihood improvement strategy. Farmers can exploit the potential domestic and export markets to increase their incomes. Therefore, the primary objective of the study was to examine the smallholder beef cattle farmers' sales decisions. Leveraging this appeal, the study specifically analyzed the determinants of market participation of smallholder beef cattle farmers, and further assessed the determinants of the level of marketing participation among the farmers.

## 2. Literature Review

### 2.1. Overview of Beef Cattle Farming

Agriculture in Eswatini is generally classified according to the dual land tenure system: Title Deed Land (TDL) and Swazi Nation Land (SNL) [20]. TDL is a freehold land system, where farmers practice specialized commercial farming on large-scale estates. The farmers are therefore, referred to as large-scale commercial farmers. On the other hand, SNL is land held in trust for the nation by the king. Traditional chiefs govern, on behalf of the king, and allocate land for settlement, crop

production and communal livestock grazing to the people. The average land size for each family is about 2.5 ha, [21], although this has since decreased due to population explosion. Farmers on SNL are generally referred to as smallholder farmers, a definition adopted for this study. Beef cattle farming on SNL is predominantly semi-subsistence farming, and beef cattle dominate the livestock sector [10] (see Figure 1).

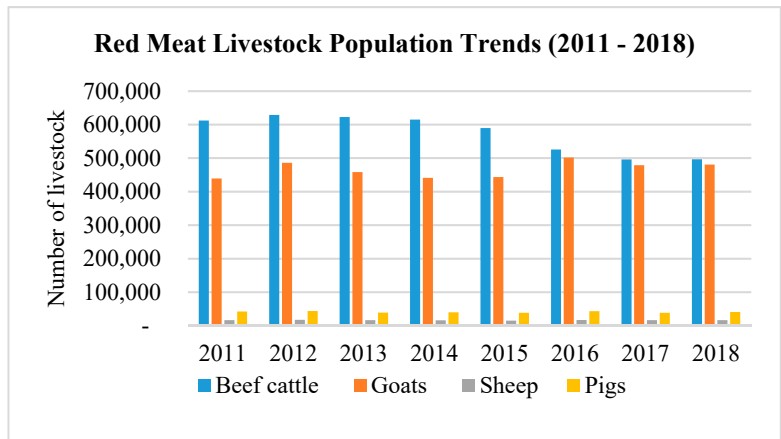

**Figure 1.** A graphical presentation of the number of red meat livestock in Eswatini from 2011 – 2018 Source: Adapted from [7,22–28].

In Eswatini, beef cattle farming is a traditional way of life and beef cattle are predominantly raised by smallholder farmers on SNL [29], under the free-range production system. The traditional Nguni breed is common among farmers and crossbreeding with exotic breeds, to improve cattle quality, is becoming popular. The national government constructs and maintains communal dip tanks, where farmers are required by law to take their cattle for tick prophylactic control and other livestock health issues. Trained veterinary assistants (VAs) provide relevant extension services and further serve as a link between the national livestock veterinary department and each community. Blood smears from slaughtered cattle or dead cattle are taken through the VAs to regional livestock health centers for diagnoses necessary for controlling livestock diseases. Although this is a postmortem analysis approach, it serves as means to ensure national food safety. Farmers can also visit the public livestock health centers for further assistance on issues of livestock health. However, private institutions and experts are available for further help at a cost.

Continuous communal grazing, which lacks pasture management and control of stocking rates, is practiced on unenclosed grasslands. This often leads to overstocking, which in turn, causes pasture and land degradation. The recent livestock census [28] indicated that 48,595 (98.52%) smallholder farmers kept a population of 496,610 (90%) beef cattle on SNL, translating to 48% of the red meat livestock population in the country. Recent statistics further indicate that 63% of the cattle slaughtered at the major export abattoir originated from SNL, compared to the 37% from TDL. This reveals the centrality of smallholder farmers in the beef industry. Therefore, this study focuses on SNL smallholder beef cattle farmers.

### 2.2. Cattle Marketing Dynamics

Since the cattle subsector in the study area lacks a structured marketing system, without regular physical marketplaces, market is defined as a virtual place or arrangement where buyers and sellers meet to complete cattle sale transactions. This is similar to the unorganized weak and indeterminate market system described by Mailu et al. [30] for indigenous birds in Kenya. Farmers practice distress sales to meet immediate cash needs, without evaluating market trends and signals for production and market participation. Cattle sales are generally spot sales, agreed and transacted through face-to-face bargaining. The use of scales and other apparatus to weigh cattle during the sale is not common, except

for transactions that involve large beef processor buyers. Pricing is done through eye-judgment based on the cattle's physical qualities.

Most commonly, buyers visit dip tanks scouting for cattle under sale and VAs are key informational resources that link farmers to potential buyers. More often than not, these buyers are representatives of beef processors and butchers who purchase cattle for beef production. In this case, market imperfections due to lack of market information impose a weak bargaining position for farmers, [31], forcing them to succumb to low prices. This has propagated a wide outcry, which could be a possible barrier to active market participation.

Apart from dip tank arrangements, farmers practice word-of-mouth advertisement within and without the community to find potential buyers. In this case, potential buyers are individual persons who buy cattle, mainly for reasons such as traditional ceremonies, home slaughters, honoring dignitaries, and so on. Such buyers are often one-time-buyers who lack bargaining experience in cattle marketing. This allows farmers to gain bargaining power; given a chance, farmers prefer such buyers for they can successfully negotiate for better prices compared to the processors and butchers' representatives at dip tanks. The challenge with this market channel is that finding customers could prove difficult, considering the nature of distress.

When push comes to shove, farmers visit or contact processors and butcheries to arrange for cattle sales. If a transaction agreement is reached, the buyers travel to pick up the cattle from the farm. In this case, farmers do not pay the transport cost, but chances are such costs are accounted for in the prices offered.

*2.3. Concept of Market Participation*

Market participation is the act by which farmers enter into agricultural markets to exchange their produce for cash. It is synonymous with commercialization, which refers to a progressive transformation of semi-subsistence farming to commercial farming [32]. This transformation is governed by profit maximization that guides production and input decisions, and the adoption of input and output market inter-linkages that enhance production and marketing processes [33]. Market participation is the first hurdle encountered by farmers, deciding whether or not to participate in cattle markets. It is succeeded by the decision on the quantity of cattle offered for sale. In literature, this second hurdle is referred to as the level, intensity, or extent of market participation. Therefore, these terms will be used interchangeably throughout this paper.

Since economic development debates and policies are market-centric [34], smallholder farmers' market participation is a fundamental tool for economic development in agriculture-based economies [12]. Coupled with enabling policy frameworks, commercialized agriculture propagates income, productivity and employment growth [35–38]. It further stimulates the establishment of infrastructure and other development-related activities, which benefits the general development of rural areas.

Broadly, in the market economy and economic development, vigorous agricultural market systems yield welfare gains for farmers through comparative advantage and increased total factor productivity growth [39]. This distributes livestock products to less productive areas, thus promoting rural and regional development [13]. By way of extension, market participation further improves smallholder farmers' welfare by enforcing market competition that reduces production cost, thereby inducing a plummet in the price of food items [40]. This, in turn, increases farmers' purchasing power, promoting the reallocation of household funds to diversified production for food and income security.

Literature has identified several determinants of market participation. Apart from price, transaction cost is a broad concept that encapsulates a variety of factors that inhibit market participation [41]. The work done by Key et al. [42] presents two broad categories of transaction costs: fixed and proportional (variable) transaction costs. Fixed transaction costs (FTCs) remain the same despite the number of cattle sold [43]. This includes search costs for customers, negotiation and bargaining costs, and screening, enforcement, and supervision costs. On the other hand, proportional

transaction costs (PTCs) involve the per-unit costs of accessing the market due to transportation and market imperfections, thereby increasing transaction costs such that market participation becomes unprofitable for some farmers. Considering this description, transaction costs are therefore said to be observable or unobservable [41], a definition we adopt for our approach. These transaction costs are thus dependent on commodity and household-specific factors, which are in turn explained by vectors of household specific characteristics, household productive assets, and public goods and services [33].

*2.4. Role of Market Participation in Sustainability*

Livestock production systems must adhere to the social demand of meeting current needs without deteriorating the ability of natural resources to meet future needs [44]. According to Lebacq et al. [45], sustainable livestock systems embrace three pillars, namely, economic sustainability, environmental sustainability and social sustainability. First, communal beef cattle farming should be economically viable to the farmer, ensuring profitability. In the case of smallholder beef cattle farming in Eswatini, where the notion of wealth storage outweighs income generation, farmers often lose livestock through deaths in recurrent droughts. Furthermore, cattle are often sold at old age, beyond their stage of peak market value. Such losses are aggravated by the low levels of market participation, thus defeating the principle of sustainable business management. The direct implication of such an economically unsustainable practice undermines the regenerative capacity of the subsector, especially after droughts, [46]. Therefore, research on smallholder market participation is paramount for sustainable rural and national economic growth [47].

Second, low market participation among beef cattle farmers contributes to global challenges of soil erosion and land degradation [48], especially in Eswatini where there is lack of communal pasture management. In a study on land degradation in Eswatini, [9], beef cattle farming was listed as one human-sourced cause of land degradation. This has induced an 18% increase in the Swati population living in degraded areas during the period 2000–2010, and the annual cost of land degradation was estimated at $100 million United States Dollar (USD) (2.9% of gross domestic product (GDP)) [49]. At the center of this dilemma is the traditional attitude of seeking prestige and honor through accumulation of cattle, thus inflating stocking rates. In turn, overstocking aggravates pasture and land degradation [10], deteriorating the ability of grazing resources to provide for future needs. Therefore, promoting market participation is a practical strategy for establishing a socially sustainable cattle production system [50,51].

In this study, we therefore argue that market participation is an indispensable part of a sustainable cattle production system in rural areas. It is a primary human-resource attitude central in the discussion on sustainable production. In our study area, the adoption of alternative farming systems such as zero grazing has been very low and legislative controls on stocking rates contradict with the traditional patronage governance system and the cultural social organization and lifestyle of the Swati people [52]. Furthermore, government-aided programs on promoting offtake from SNL have struggled due to their small capacity and insufficient funds. Therefore, active market participation remains a plausible strategy towards sustainable production to complement existing policies and development frameworks for this unique nation [53].

An advanced view places market participation as a critical pragmatic mitigation strategy for climate change, to prevent farm losses due to recurrent droughts. In 2016, the El Niño drought claimed more than 63,000 herd of cattle, [54], resulting in losses for farmers. Hence, market participation can serve as an operational strategy to minimize such livestock losses through enhancing offtake rates to maximize economic and environmental benefits for rural farmers.

Acknowledging the earlier research work that incorporated sustainability into marketing [55], literature, as far as we are aware, has rarely presented farm–firm market participation as a practical component of the sustainability discussion. Several studies have addressed marketing from the perspectives of production and processing [56–60], but not as a standalone, connective component

within the production–consumption continuum. Therefore, our study provides for this gap in literature on sustainable production and advances the discussion of marketing for sustainability.

## 3. Materials and Methods

### 3.1. Study Area

The Kingdom of Eswatini is a 17,364 km$^2$ country landlocked between South Africa and Mozambique, with a population of about 1.2 million. The country is classified as a lower middle-income country with a gross domestic product per capita of $3224.39 USD, where 62.1% of the people live below the poverty line of $3.20 USD per person per day [61]. About 77% of the population depends on subsistence agriculture in rural areas [62].

The country is divided into four administrative districts: Hhohho, Lubombo, Manzini and Shiselweni. Generally, the country experiences varied climate, subtropical to near temperate, over the four agro-ecological zones (Highveld, Middleveld, Lowveld and Lubombo Plateau). The Highveld experiences a wet and cool climate, receiving an average of 700–1550 mm of rainfall [63]. The combination of high rainfall amounts and slightly acidic soils promotes the growth of a wide variety of grasses called sourvelds due to their poor quality, thus poor quality livestock. The Middleveld and Lubombo Plateau share most of the same climatic characteristics, receiving a good amount of rainfall (550–850 mm) and warmer temperatures. The fertile soils and good climatic conditions allow for the growth of highly palatable grasses good for high-quality cattle. The Lowveld receives very low levels of rainfall (400–550 mm) and is highly susceptible to recurrent drought, thus it is not suitable for cattle production.

### 3.2. Sampling and Data Collection

The study targeted smallholder beef cattle farmers ($N$ = 48,595) and a sample of 397 farmers was determined through the application of Slovin's formula [64] as

$$n = \frac{N}{1 + Ne^2} = \frac{48,595}{1 + 48,595(0.05)^2} = 396.734 \approx 397 \tag{1}$$

where $n$ is the size of the sample; $N$ is the target population; $e$ is the error tolerance level (0.05).

A three-stage stratified simple random sampling technique was utilized to draw respondents from the population. First, farmers were stratified according to the four districts (Hhohho, Lubombo, Manzini and Shiselweni). The percentage proportion was then utilized to determine the number of farmers to be selected from each stratum (see Table 1). The second stage involved grouping the farmers into market participants ($s_1$) and non-participants ($s_2$), ensuring that both subsamples were well represented for analysis ($s_1$ = 200, $s_2$ = 197). Then simple random sampling was applied in the third stage to select the respondents of the study.

**Table 1.** Sample size.

| District | Population | Percentage Proportion | Sample |
|---|---|---|---|
| Hhohho | 13,290 | 27.35 | 109 |
| Lubombo | 9649 | 19.86 | 79 |
| Manzini | 14,520 | 29.88 | 118 |
| Shiselweni | 11,136 | 22.92 | 91 |
| Total | 48,595 | 100 | 397 |

Data were collected through personal interviews, guided by a structured questionnaire, between September and December 2018. A structured interview entails verbally administering predetermined questions with an opportunity to clarify certain questions in case literacy and numeracy problems exist among respondents [65]. Therefore, the method best suited our study since the likelihood

of the prevalence of such problems was high. Moreover, the use of structured questionnaires to guide interviews ensures completeness in data provided in an organized way with a high response rate [66]. Our experience in data collection in the study area also justified the use of structured personal interviews, since traditionally, farmers prefer a face-to-face conversation that begins with explaining the purpose and importance of the required data.

The questionnaire was divided into four parts: Part I focused on production-related information (number of cattle, number of slaughters, feed, labor and relevant costs). The first question required the farmer to indicate the cattle breed type. Data collection on number of cattle, home slaughters and cattle deaths was organized in tabular format according to cattle categories, estimated values and relevant reasons. Collection of data on feed and inputs was organized in table formats specifying the name, quantity and cost of input. Part II was dedicated to collect data on public goods and services: access to credit, frequency of extension and veterinary visits and pasture availability. Part III focused on market factors: market channel, number of cattle sold according categories, distance to buyer and relevant marketing costs. Part IV was specifically for socio-economic farmer-household characteristics: gender, education, experience, marital status, household size, off-farm employment, amount of monthly non-farm income and membership to farmers' association.

The interviews were conducted in a systematic manner to maximize data collection output. The first phase involved an introduction of the interviewer with a firm handshake and small talk on the weather and current issues; a traditional introductory conversation kneeling on the ground. The second phase involved a brief introduction of the purpose of the interview and importance of the data collected, seeking permission to continue with the interview. Once permission was granted, the actual interview proceeded through questioning and responses were recorded directly into the questionnaire (third phase). Fourth, the respondents were encouraged to ask questions on the subject matter and other agricultural issues in order to provide useful information and advice where possible. Finally, interviews were concluded with a verification of contact information and humble valedictions.

### 3.3. Conceptual Framework

The theoretical basis for farmers' sales decisions is grounded on the agricultural household model [67,68], in which a farming household utility is maximized from a basket of produced and/or purchased goods, subject to the income constraint imposed by earning from the mix of household production, sales and off-farm employment. The household is faced with the first decision—whether to enter the market or not (participation decision)—where the value of one represents a household that participates in cattle marketing, and zero indicates otherwise. The second decision is based on the quantity of cattle offered for sale (quantity decision), conditional to the participation decision.

In the context of Eswatini, smallholder beef cattle farmers are confronted with numerous market imperfections, hence, we follow Olwande et al. [33], who invoked the non-separable household model for market participation [30,37]. In this regard, two features explain the stylized model of household market-participation behavior. First, market participation imposes different transaction costs on farming households, inducing non-uniform market behavior [42]. Second, spatial differences induce varying commerce costs on household market participation due to different geographic settings. Together with market price, these key features guide rational farming households' decisions [33] on whether to enter into the market or not. Households that eventually decide to participate, further self-select market outlets that provide maximum incentives, thereby deciding on the intensity of participation. The implication of this behavior is on structural patterns of market participation that induce substantive implications for agricultural development policy required to stimulate agricultural productivity growth and rural poverty alleviation [37].

Based on Olwande et al. [33] and Barrett [37], we designate the level of market participation ($Q_i^{CS}$ a vector of cattle sold subject to *k*, an indicator variable equal to one for market participation, zero indicates otherwise) to be a function of observed market prices (P) and the determinants of transaction cost. Transaction cost, the observable and unobservable costs imbedded in arranging and completing cattle

sales [41], depends on the vectors of farmer-household characteristics (FH), market-related factors (M) and public good and services (PG). According to Barrett [37], the farmer-specific, household-specific and location-specific characteristics are a theoretical source of transaction costs that affect the marketing of agricultural produce. Differences in the determinants of transaction costs underscore the heterogeneity in market participation among cattle farmers. Specific farmer-household features such as education, age, experience, gender and so on, affect search costs, negotiating skills, etc. Production wise, the vector of farmer- and household-related factors account for the skills and efficiency in the use of production resources to increase marketable surplus, thus influencing market entry and the extent of market participation [69]. These variables do not only capture the role of human capital, but also the mitigation of transaction cost through increased ability to obtain market information and establish trading networks [41]. Furthermore, household characteristics such as household size affect the availability of labor for the production of marketable surplus, thereby, influencing sales decisions.

Since cattle sales include the selection of a suitable herd, we added a vector of cattle-related factors (C) such as number of cows, steers, heifers, and calving and calf mortality rates to capture the availability of marketable surplus and cattle reproductive and replacement potential. Basically, this vector embraces direct production shifters that generate marketable surplus, which are often neglected in market participation studies [41]. Cows, heifers, calving rate and calf mortality rate reflect reproductive and replacement capacity of the herd, which motivates farmers to engage in cattle marketing. Bahta and Bauer [38] included livestock births as means of capturing the influence of herd structure on livestock marketing decisions, and herd structure was found to be statistically significant in this regard. Notably, Mailu et al. [30] also included flock characteristics in their analysis of price effect on market participation decisions by indigenous poultry farmers in Kenya.

Theoretically, access to price and market information promotes market participation by reducing transaction cost [42]. When accurate market information is readily available, farmers are able to access market channels with high market incentives, thus promoting market participation and the extent of market participation. Therefore, market-related factors (M) are critical for the marketing of agricultural products.

PG represents a vector of public good and services provided by the local and national governments for farmers. The relevant public goods and services for our analysis include access to credit, pasture availability and extension services [33,37]. Availability of sufficient pastures provides sufficient good quality grazing for cattle, thus promoting the production of marketable surplus necessary for market entry. Extension services enhance market participation through skills and knowledge impartation. Credit access depends on the institutional environment and the terms and conditions set by the government and financial institutions, which affect the availability of farm credit to farmers. Farm credit is critical for the production of marketable surplus, thus influencing market participation.

Therefore, the reduced form of our level of market participation model, subject to the decision of market entry, is specified as

$$Q_i^{cs} = Q_i^{cs}(\text{k, P, FH, C, M, PG}) \tag{2}$$

where $Q^{CS}$ is the number of cattle sold by the *i*th farmer; k is the decision to participate in the market (1 = participates, 0 = otherwise); P is a vector of observed market prices; FH is the vectors of farmer and household characteristics; C is the vector of cattle-related factors; M is the vector of market-related factors; and PG is the vector of public goods and services.

### 3.4. Analytical Framework

The study employed two statistical approaches for data analysis. First, descriptive statistics (means, standard deviation, frequencies and percentages) and inferential statistics (independent *t*-test and chi-squared test) were applied to describe the variables used for analysis. The independent *t*-test was applied to determine statistically significant differences between market participants and non-participants with regards to continuous variables. The chi-squared test was used for determining statistically significant differences between the subsamples with regards to categorical variables.

The second analysis approach involved econometric analysis to examine the determinants of market participation and the level of market participation.

The analysis of market participation entails a situation where at each observation the event may or may not occur. An occurrence (market participation) is associated with a continuous non-negative random variable, while a non-occurrence (not participating) yields a variable with zero value [70]. Such a scenario presents a limited dependent-variable [71] modeling problem where the lower bound of the variable, zero value, occurs in a considerable number of observations. The occurrence of the event allows a continuous distribution over the positive values, but a "pile up" at zero exists (due to non-occurrence), which is a corner solution for the participation problem [72]. Such common cases, in social science, render invalid the use of the usual regression model and require models that can handle binary endogenous variables.

The common approaches of modeling such situations include the Tobit, Heckman and double-hurdle models [73]. Several studies have applied the Tobit model to address farmers' market participatory decisions [47,69,74,75], but the major drawback of this approach is that it imposes a restriction that both sales decisions are simultaneously influenced by the same set of explanatory variables [76]. Since we assume, in this study, that the decisions on market participation and level of market participation are influenced by different sets of independent variables, the Tobit model is not recognized. It is also argued that the model yields biased parameter estimates [77] and recent studies have stressed the inadequacy of the Tobit, proposing the use of less restrictive alternative approaches—the Heckman model [78] and Cragg's double-hurdle model [70]. These alternative two-stage models are relevant to our study for they assume that separate vectors of independent variables influence the farmer's sales decisions.

The Heckman applies the probit model in the first stage (selection model) for the participation decision and the ordinary least squares regression model (outcome model) for the quantity decision. The zeros in the selection model are treated as cases of unobserved or missing data [78]. The selectivity term, the inverse Mills ratio computed in the selection model, is incorporated in the outcome equation to rectify sample-selection bias [79]. This ensures consistent and efficient asymptotic parameter estimates [78]. The model is popular in market participation literature, applied in dairy [80], poultry [81], small-ruminants [77] and beef cattle [82]. However, the model is best suitable for non-random samples [73] and is deficient when the normality assumption is violated [83]. For these reasons, this approach was not considered for our analysis.

The double hurdle is a less restrictive variant of the Heckman and is best suited for samples drawn through random probabilistic sampling procedures [73]. Therefore, the double hurdle was adopted for the analysis of our randomly selected sample data. The model is a generalization of the Tobit, where two separate stochastic processes determine the participation and quantity decisions. The notable difference between the two-stage models is based on the assumption of the Heckman, that non-participants will not participate under any circumstance [83]. Contrary, the double hurdle assumes that the decision not to participate is a deliberate choice [84], thus the zeros from non-participants are considered as corners solution in the utility maximizing model [85]. This model further curbs bias in the continuous second tier dependent variable by linking a value to the piled-up data, thus maintaining all the data within the sample. The model is also flexible, assuming that there are no restrictions regarding the components of independent variables in each estimation stage. The model has been used in market participation literature. Notably, Ndoro et al. [12] applied the model in analyzing cattle commercialization in South Africa. The model has also been applied in dairy [83], and crop marketing [84,86].

Further tests for appropriateness between the Tobit and the double-hurdle models were ascertained by the Wald chi-squared test and Akakie's information criterion [85]. The double hurdle was found to fit better with the analysis than the Tobit, and was thus adopted for this study.

### 3.5. Estimation Strategy

The double hurdle model is more flexible than the Tobit and allows the participation and quantity sales decisions to be determined separately [87]. The model requires a joint application of the probit and truncated regression models, sequentially or simultaneously [85]. The theoretical basis of the double-hurdle estimation framework by Cragg [70] is grounded on the probit model where the probability of market participation at observation $t$, $p(E_t)$, is given by

$$p(E_t) = \int_{-\infty}^{X_i'\beta} (2\pi)^{-\frac{1}{2}} \exp\left\{-z^2/2\right\} dz \tag{3}$$

where $X_t$ is a K $\times$ 1 vector of exogenous variables at observation $t$ and $\beta$ represents a vector of parameter estimates. Then the cumulative unit normal distribution is designated as

$$C(z) = \int_{-\infty}^{z} (2\pi)^{-\frac{1}{2}} \exp\left\{-t^2/2\right\} dt \tag{4}$$

The probit model estimates the probability of a farmer to participate in cattle marketing (first sales decision). The second sales decision—number of cattle offered for sale—occurs when favorable circumstances (search, information and transaction costs) prevail to allow the transaction to be completed [88]. This non-negative quantity decision can only be measured for non-zero values in the first decision, thus estimated by the truncated regression [76]. Therefore, the double-hurdle two-equation framework [89,90] is presented as

$$MP_i^* = Z_i'\alpha + \varepsilon_i \text{ Participation decision} Q_i^{CS**} = X_i'\beta + u_i \text{ Quantity decision} \begin{pmatrix} \varepsilon_i \\ u_i \end{pmatrix} \sim N\left[\begin{pmatrix} 0 \\ 0 \end{pmatrix}, \begin{pmatrix} 1 & 0 \\ 1 & \sigma^2 \end{pmatrix}\right] \tag{5}$$

where $MP_i^*$ is the latent variable for the binary dependent variable taking a value of one for market participation and zero indicates otherwise. $Q_i^{CS**}$ is the latent variable reflecting the number of cattle sold. $Z_i$, $\alpha'$ and $\varepsilon_i$ represent vectors of explanatory variables, parameter estimates and the error term for the market participation decision. Likewise, $X_i$, $\beta'$ and $u_i$ represent vectors of explanatory variables, parameter estimates and the error term for the level of market participation. Since an individual farmer is involved in both sales decisions, the error terms are assumed to be independently and normally distributed, thus the first hurdle corresponds to a probit model [90].

The binary dependent variable of the participation decision in Equation (5) is defined by

$$MP_i = 1, \text{ if } MP_i^* > 0 \quad MP_i = 0, \text{ if } MP_i^* \leq 0 \tag{6}$$

and decisions on the level of market participation is defined by

$$Q_i^{CS*} = \max\left(Q_i^{**}, 0\right). \tag{7}$$

The observed variable, $Q_i^{CS}$ (normally presented as $y_i$ in literature) is determined as

$$Q_i^{cs} = MP_i \cdot Q_i^{CS*} \tag{8}$$

and log-likelihood function for the double hurdle is

$$\text{LogL} = \sum_0 ln\left[1 - \Phi\left(Z_i'\alpha\right)\Phi\left(\frac{X_i'\beta}{\sigma}\right)\right] + \sum_+ ln\left[\Phi\left(Z_i'\alpha\right)\frac{1}{\sigma}\Phi\left(\frac{Y_i - X_i'\beta}{\sigma}\right)\right]. \tag{9}$$

The impact of the explanatory variables on the dependent variables is assessed by the analysis of marginal effect, thus, the unconditional mean is decomposed into the effect on the probability of

participating in cattle marketing and the effect on the conditional intensity of market participation. Following Ndoro et al. [12], when these components are differentiated with respect to each explanatory variable, the unconditional mean becomes

$$\mathrm{E}[Q|X_i] = P(Q_i > 0)\cdot E(Q_i|Y_i > 0). \tag{10}$$

Then the probabilities of market participation and expected number of beef cattle sold, conditional on the first decision to participate are presented as

$$\mathrm{P}(Q_i > 0) = \Phi\big(Z'_i\alpha\big)\Phi\left[\frac{X'_i\beta}{\sigma}\right] \tag{11}$$

and

$$\mathrm{E}(Q_i|Q_i > 0) = \Phi\left(\frac{X'_i\beta}{\sigma_i}\right)^{-1}\int_0^\infty\left(\frac{Q_i}{\sigma_i\sqrt[i]{1 + \theta^2 Y_i^2}}\Phi\left(\frac{T(\theta Q_i) - X'_i\beta}{\sigma_i}\right)\right)dY_i. \tag{12}$$

Based on the above-described econometric framework, the probit model for the first sales decision on market participation (MP) was specified as

$$
\begin{aligned}
\Pr(\mathrm{MP} = 1) = \quad & \alpha_0 + \alpha_1 Education + \alpha_2 Age + \alpha_3 Gender + \alpha_4 Experience + \alpha_5 Location \\
& + \alpha_6 MaritalStatus + \alpha_7 Householdsize + \alpha_8 OffFarmEmployment \\
& + \alpha_9 Expenses + \alpha_{10} Herdsize + \alpha_{11} PregnantCows + \alpha_{12} Slaughters \\
& + \alpha_{13} SteerHeiferRatio + \alpha_{14} Breedtype + \alpha_{15} CalvingRate \\
& + \alpha_{16} CreditAccess + \alpha_{17} PastureAvailability + \alpha_{18} Extension + \varepsilon_i.
\end{aligned}
\tag{13}
$$

The parameter estimates ($\alpha$) accord the signs of the partial effects of the explanatory variables, $Z_i$, on the probability of the outcome variable. Then their marginal effects are used for evaluating the effect of each independent variable on the outcome variable [91].

The truncated regression model for determinants of the level of market participation (second sales decision) was specified as

$$
\begin{aligned}
\text{No. cattle sold} = \quad & \beta_0 + \beta_1 Education + \beta_2 Gender + \beta_3 Expereince + \beta_4 HouseholdSize \\
& + \beta_5 MaritalStatus + \beta_6 Association + \beta_7 Laborers + \beta_8 NonFarmIncome \\
& + \beta_9 Expenses + \beta_{10} CreditAccess + \beta_{11} Extension + \beta_{12} PastureAvailability \\
& + \beta_{13} Breedtype + \beta_{14} Cows + \beta_{15} Heifers + \beta_{16} Steers + \beta_{17} CalvingRate \\
& + \beta_{18} CalfMortaliyrate + \beta_{19} MarketPrice + \beta_{20} MarketDistance \\
& + \beta_{21} MarketInformation + \beta_{22} MarketChannel + \beta_{23} SaleDuration + \varepsilon_i.
\end{aligned}
\tag{14}
$$

The independent variables used in the double-hurdle analysis are presented in Table 2, with their definitions and hypothesized signs. The assumption of the econometric estimation procedure required that multicollinearity be controlled in order to generate non-biased parameter estimates. For continuous variables, the bivariate correlation matrix and variance inflation factor (VIF) were applied to identify and eliminate the collinear variable [92]. Contingency table analysis was performed using chi-squared, contingency coefficients [84] and lambda to identify sources of multicollinearity among categorical independent variables. In addition, the STATA 15 statistical package was used to suppress collinear variables during analysis. The robust standard error for estimated parameters was used to curb potential heteroscedasticity within the error term.

**Table 2.** Definition and expected signs for explanatory variables used in the analysis.

| Variable | Definition | Measurement | Expected Sign | |
|---|---|---|---|---|
| | | | Market Participation | Level of Participation |
| **Farmer-Household Characteristics** | | | | |
| Employment | Off-farm employment | 0 = No; 1 = Yes | +/- | |
| Age | Farmers' age | Years | +/- | |
| Ecological zone | Climatic region | 0 = LV; 1 = MV; 2 = HV; 3 = L | + | |
| Education | Formal schooling | Years | + | + |
| Gender | Sex of the farmer | 0 = Female; 1 = Male | + | + |
| Experience | Farming experience | Years | + | + |
| Household size | Individuals in a household | Number | +/- | +/- |
| Marital status | Married or not | 0 = Single; 1 = Married; 2 = Widowed | + | + |
| Association | Farmers' association membership | 1 = Yes; 0 = No | | + |
| Laborers | No. of laborers | Number | | + |
| Non-farm income | Monthly off-farm income in Emalangeni | 0 < E5000; 1 ≥ E5000 | | +/- |
| **Cattle-Related Factors** | | | | |
| Herd size | Total cattle kept | Number | + | |
| Steer–heifer ratio | Proportion of steers and heifers to herd size | Ratio | +/- | |
| Pregnant cow | No. of pregnant cows | Number | + | |
| Slaughters | No. of cattle used for home slaughters | Number | +/- | |
| Calving rate | Proportion of weaned calves to no. of pregnant cows | Ratio | + | + |
| Breed type | Type of cattle kept | 0 = Nguni; 1 = Crosses | + | + |
| Cows | Female cattle that have calved | Number | | + |
| Heifers | Young females that have not calved | Number | | + |
| Steers | Young neutered males | Number | | +/- |
| Calf mortality rate | Proportion of dead calves to no. of pregnant cows | Ratio | | - |
| **Market-Related Factors** | | | | |
| Expenses | Total expenses | Emalangeni | + | + |
| Market price | Average price of cattle sold | Emalangeni | | + |
| Market distance | Distance from farm to buyer's location | Kilometer | | +/- |
| Market channel | Buyer of cattle | 1 = Processor; 2 = Individual 3 = Combination | | + |
| Market information | Source of market information | 1 = Informal; 2 = Formal | | + |
| Sale duration | Days to successful cattle sale | Days | | +/- |
| **Public Goods and Services** | | | | |
| Credit access | Access to farm credit | 1 = Yes; 0 = No | + | + |
| Pasture availability | Perception on pasture availability | 0 = Insufficient; 1 = Sufficient | + | + |
| Extension | Extension visits | Number | + | + |

Note: LV = Lowveld, MV = Middleveld, HV = Highveld, L = Lubombo Plateau.

*3.6. Definition of Variables and a Priori Expectations*

The binary dependent variable for estimating the probability of market participation in the probit regression model takes the value of one, for a farmer that sold cattle, and zero for a farmer that did not sell. The outcome variable for the truncated regression model is the number of cattle sold, subject to the first decision to sell.

3.6.1. Farmer-Household Characteristics

The farmer-household characteristics capture the capability of the farmer to employ production resources and manage marketing processes. Education enhances skills application and information utilization required for cattle marketing [83]. The variable is also an indicator for the adoption of

innovations and new technology necessary for increasing farm productivity. Based on the significant positive effect of education on market participation, [82], a positive association between the variable and the sales decisions is envisaged.

Experience is a continuous variable measuring the number of years the farmer has been involved in cattle farming. Since the cattle marketing system in the study area is unorganized, experience captures the effect of social networks and links that accrue over time to enhance the search for trading partners [93]. The variable is expected to bear a positive effect on farmers' sales decisions.

Age was measured as a continuous variable representing how old the farmer was as of their last birthday. Considering that beef cattle production is a traditional way of life in the study area, age captures the level of farmer stereotypes towards market participation. Older farmers use less productive traditional methods and are less willing to engage in cattle marketing. According to Egbetokun and Omonona [86] and Kgosikoma and Malope [94], age is negatively associated with farmers' sales decisions, whereas Randela et al. [95] found a positive relationship for this variable with market participation. Therefore, an indeterminate relationship is hypothesized.

Gender, a dummy variable, is expected to affect farmers' sales decisions in the study area. Female farmers are often more concerned about household self-sufficiency than their male counterparts. A study by Abeykoon et al. [81] on farmers' market participation in indigenous poultry revealed that market participation was positively and significantly associated with male farmers. In their study, Farinde and Ajayi [96] they found a low sample proportion of females (25%) to be involved in livestock marketing activities, the lowest compared to other livestock management activities. Therefore, a positive relationship between gender and sales decisions is expected with respect to male farmers.

Marital status was measured as a three-level categorical variable. The variable captures the effect of support in decision-making and livestock management that the farmer receives. Married farmers have a wider base for decision-making, thus marital status is posited to have a positive impact on sales decisions with regards to a married marital status.

Household size was measured as a continuous variable that reflects the number of people in a household. Larger households have the potential of more family labor required for production and marketing functions. Contrary, larger households may indicate pressure on livestock resources to generate funds for livelihood needs such as school fees and so on. Therefore, an indeterminate relation between the variable and sales decisions is hypothesized. A positive association would be indicative of efficient use of family labor to promote market participation and intensity of market participation. However, a negative relationship will reflect inefficient use of labor and/or livelihood pressure that depletes the cattle herd to reduce the farmer's involvement in cattle marketing [39,97].

Cattle farming requires capital for expenses, thus a positive impact on sales decisions is presumed for off-farm employment and non-farm income. The variables were incorporated as dummy variables capturing the financial ability of the farmers to sustain cattle farming. Otherwise, a negative relationship is expected when farmers expend more time on non-farm employment over farm activities, and do not reinvest non-farm income into farm production and marketing activities [40,41].

The ecological zone, based on climatic conditions, determines the quality of grass for cattle production. This variable is a location-specific feature that affects involvement and extent of involvement in agricultural markets [84] through the effect of competitive advantage. Agro-ecological zones that receive sufficient rainfall produce palatable grasses that improve livestock quality, thereby, improving cattle quality that warrants high market value. This in turn increases market incentives for market participants, thus a positive relationship is anticipated between regions that receive sufficient rainfall (Middleveld, Highveld and Lubombo Plateau) and market participation.

Farmers' associations were used as a dummy variable to capture whether a farmer is a member of an association that is involved in production and/or marketing of agricultural products. Farmers' associations, also called cooperatives, are useful in information sharing, resource mobilization and extension services, thus the variable is hypothesized to be positively related to the cattle sale decisions.

In a study on smallholder commercialization in Ethiopia, Abafita et al. [39] found that cooperative membership promotes cattle marketing.

Labor is a continuous variable that measures the exact number of persons responsible for the day-to-day care of cattle. Farmers that have several people to look after the cattle bear the ability of managing larger herd sizes, thus increasing the propensity for market involvement.

### 3.6.2. Cattle-Related Factors

For purposes of this paper, inclusion of cattle-related variables is a unique innovation meant to capture the effect of herd dynamics in encouraging farmers to engage in cattle marketing. In cattle marketing, unlike crop marketing, the selection and grading process is not for preparing produce for market, but a critical decision that acts as a mini-hurdle for farmers to enter the market. Farmers are obliged to select cattle to maximize income yet not hamper the multiplicative ability of the herd.

Herd size depicts the total number of cattle kept by a farmer at a particular time. Practically, a large herd size implies a more marketable surplus readily available for sale, which is a motivating factor for market entry. In light of population growth in the context of declining land–person ratio [98], agricultural development through critical mass production and marketing does not necessarily impose the threat of smallholder farmer marginalization. Therefore, the future of smallholder farmers, as of now, is bright. Hence some studies on market participation have suggested herd size expansion to stimulate active market participation by smallholder farmers [1,30]. Therefore, the variable is hypothesized to be positively related to market participation.

The numbers of cows is a crucial herd dynamic that captures the reproductive potential of the herd, which promotes high levels of market participation. A high number of cows entrusts belief to the farmers that the herd size will increase through births, thus the availability of sales replacements. Therefore, a farmer that has more cows is anticipated to exhibit high market intensity; hence, a positive effect is envisaged.

The number of heifers and steers in the herd is another herd feature that influences sales decisions. First, cow and bull calves provide a pool for the selection of replacement-breeding stock. Selected heifers and bull calves transform culled breeding stock to marketable surplus and non-selected bull claves are castrated to produce more marketable surplus. Steers, in particular, are highly saleable further increasing the extent of market participation. Similarly, the steer–heifer ratio depicts cow replacement potential and availability of marketable surplus. However, high steer–heifer ratios could reflect low market participation for stereotyped farmers that hold on to cattle for long periods. Hence, an indeterminate relationship is expected for this variable with market participation.

The calving rate depends on the number of pregnant cows. The variable was measured as the proportion of weaned calves to the number of pregnant cows at the beginning of the breeding season. Calving rate and number of pregnant cows capture farm management efficiency in the production process. In their assessment of determinants of market participation within the South African small-scale livestock sector, Bahta and Bauer [99] found livestock births to have a positively and statistically significant effect on livestock marketing. Therefore, a positive association is hypothesized between these variables and the sales decisions. On the other hand, calf mortality rate captures low farm efficiency in the production process, which reduces marketable surplus. Thus, a negative relationship is hypothesized with the intensity of market participation.

On another note, home slaughters are indicative of a farmer's luxurious behavior, unless they are distress slaughters. Beef cattle in the study area are basically not kept for family consumption like in crop production. The livestock are kept as a store of wealth, converted to cash through distress sales. Thus, high levels of home slaughters capture the farmer's willingness and openness to increase offtake rate. Therefore, a positive association is anticipated with market participation, but the contrary is envisaged for distress sales.

Breed type was incorporated as a dummy variable to identify the type of cattle kept by farmers. The variable captures the breeding and growth potential of the herd and the quality of cattle produced.

The native Nguni breed is a small-sized slow-growing breed with low market value. Although pure exotic breeds are scarce, due to high management cost, crossbreeding the Nguni cattle with the Brahman breed is popular. Crossbreeding is advantageous through breed and heterosis effects [100]. Therefore, keeping crossbreeds is associated with the production of high quality cattle, which increases the value of marketable surplus. Thus, a positive association between breed type and both sales decisions is envisaged regarding crossbred cattle.

### 3.6.3. Market-Related Factors

Distance to market represents the estimated distance from the farmer's location to the buyer's location in kilometers. It is the distance traveled by the buyer when scouting for cattle under sale. Literature widely reveals a negative effect of market distance on marketing activities [6]. However, other studies have found a statistically significant positive association with sales decisions, especially for farm gate marketing [94,95]. Thus, an indeterminate relationship is expected.

Market channel was used as a categorical variable to capture the effect of the different market channels on the extent of market participation. The variable was included as an innovation since different market channels offer different prices. Output price is the ultimate incentive for market participation; therefore, channels offering higher prices promote the intensity of market participation. Since cattle are transacted through face-to-face negotiations, a positive relationship is posited for market channels in which farmers have greater bargaining power.

The lack of an organized marketing structure enforces dependence on informal sources for market information. Therefore, the variable is posited to impose a positive effect on the intensity of market participation in relation to informal sources of market information.

Sales duration is another innovation that captures the impact of the length of time in making a successful cattle sale transaction. A longer sale period can accord bargaining power to sellers, thus attracting favorable prices and promoting level of market participation. However, a longer duration could also reduce the seller's bargain power because of the urgency of the sale, thereby imposing unfavorable prices that negatively affect the extent of market participation. Hence, an indeterminate effect is anticipated.

Total expenses is a composition of input market effect through the cost of inputs and services, and marketing costs (bus fare, telephone, transport and so on). In terms of the cost of soliciting inputs and services, the variable is a proxy for investment into the production and marketing processes, thus posited to have a positive relation with marketing intensity. However, a negative impact may be imposed by the marketing costs, although this component is expected to be of less influence since the study is dealing with the analysis of farm gate sales.

Output price is the ultimate incentive for sellers; therefore, a positive relation is anticipated for this variable with intensity of market participation. In assessing the determinants of the intensity (percentage) of kale sold by smallholders, Omiti et al. [40] found a positive and statistically significant association for this variable. Furthermore, Rutto et al. [1] found a similar result in their analysis of market participation for small ruminant livestock keepers.

### 3.6.4. Public Goods and Services

Access to credit was incorporated as a dummy variable aimed at capturing the effect of availability of funds for agricultural production and marketing. In an analysis of farmers' sales decisions on teff marketing, Tura et al. [84] found a significantly positive impact of credit access on market participation. Credit further enhances the acquisition of input resources, extension visits and other services for production and marketing processes. Therefore, a positive relationship is envisaged in relation to access to credit.

Pasture availability, a dummy variable, depicts the availability of grazing resources for cattle farming. Since stocking rates are not controlled in the study area, a perception by farmers that grazing resources are sufficient encourages the farmers to invest in production processes, thereby increasing

farm productivity and herd size. This further ensures the availability of marketable surplus, thus promoting cattle marketing. Therefore, a positive association with sales decisions is posited.

Frequency of extension visits is a continuous variable set to capture the support received by farmers in terms of production and marketing information. The variable was found to exert a statistically significant positive influence on market participation. Acquisition of extra extension and veterinary visits captures the investment cost incurred by a farmer. Since investment improves farm productivity, extension visits are posited to bear a positive effect on cattle sales decisions.

## 4. Results and Discussion

### 4.1. Descriptive Analysis

Means, standard deviation, frequencies and percentages (descriptive statistics) and inferential statistics of significant differences (independent *t*-test and chi-squared test) were adopted to describe and compare the variables used in the study. The descriptive comparison of categorical variables based on frequency counts and the chi-squared test is presented in Table 3. Statistically significant differences at $p < 0.01$ are revealed with respect to breed type, credit access, gender, non-farm income and marital status. The results reveal that market participants tend to keep hybrid cattle over the traditional Nguni breed ($Cross_{S1}$ = 120, $Cross_{S2}$ = 35), whereas more non-participants tend to keep the traditional breed ($Nguni_{S1}$ = 80, $Nguni_{S2}$ = 162). Among the participants, 60% raised crossbreds compared to the 18% among non-participants. However, 61% of the sample population raised the Nguni breed, proving its popularity in the study area. Since hybrid progenies are produced through crossbreeding, participants that keep crosses are at liberty to benefit from the heterosis effect that yields high quality cattle through superior hybrid vigor [100]. Therefore, the results suggest a positive relationship between cattle marketing and breed type with respect to crossbred cattle.

**Table 3.** Participants compared to non-participants based on frequencies of categorical variables.

| Variable | Participants ($s_1$ = 200) | Non-Participants ($s_2$ = 197) | $\chi^2$ |
|---|---|---|---|
| Breed type | Nguni = 80; Cross = 120 | Nguni = 162; Cross = 35 | 74.380 *** |
| Credit access | No = 158; Yes = 42 | No = 186; Yes = 11 | 20.390 *** |
| Pasture availability | Insufficient = 102; Sufficient = 98 | Insufficient = 112, Sufficient = 85 | 1.368 |
| Gender | Female = 51; Male = 149 | Female = 83; Male = 114 | 12.278 *** |
| Off-farm Employment | No = 83; Yes = 117 | No = 96; Yes = 101 | 2.096 |
| Non-farm income | <E5000 = 82; ≥E5000 = 118 | <E5000 = 124; ≥E5000 = 73 | 19.144 *** |
| Marital status | Single = 4; Married = 149; Widowed = 47 | Single = 11; Married = 111; Widowed = 75 | 15.225 *** |
| Ecological zone | Lowveld = 36; Middleveld = 107; Highveld = 52; Lubombo = 5 | Lowveld = 33; Middleveld = 115; Highveld = 48; Lubombo = 1 | 3.223 |
| Association | No = 188; Yes = 12 | No = 187; Yes = 10 | 0.162 |

Significance level: *** $p < 0.01$.

Farm credit is essential for investment in production and marketing processes, thus promoting farm productivity. Increased productivity yields marketable surplus, which in turn, increases the propensity for market entry and the extent of market participation. The results reveal that only a small proportion of the sample (13%) had access to farm credit. However, sample dynamics indicate that more farmers (21%) among the participant subsample were exposed to credit compared to the meager 6% among non-participants. This suggests a positive association between access to credit and involvement in cattle marketing.

The results indicate that males dominate beef cattle farming, making up 66% of the total sample. Further analysis reveals that out of the 34% of female farmers in the sample, only 38% engaged in cattle marketing compared to the 57% within the male population. This goes to show that given a chance

in cattle ownership and production, females will be less active in livestock market participation [96]. The results, therefore, suggest a positive effect of gender on sales decisions with respect to males.

The results further reveal that among participants, 59% earned non-farm income that is greater or equal to E5000, compared to 37% among non-participants. If reinvested into production and marketing activities, non-farm income increases productivity and marketable surplus, thereby increasing the probability of market entry and the level of market participation. This proposes a positive association between cattle marketing activities and non-farm income.

Marital status was included in the analysis as a categorical variable with three levels: single, married and widowed. More market participants were married than non-participants were ($Married_{S1}$ = 149, $Married_{S2}$ = 111) and were less likely to be widowed ($Widowed_{S1}$ = 47, $Widowed_{S2}$ = 75). This reflects a wider human resource base for decision-making and labor for participants, thus increasing marketable surplus. Furthermore, 75% of participants were married compared to 56% of non-participants. The results suggest that market participants are more exposed to a wider base of human resource for management and decision-making functions. Thus, the married marital status is expected to be positively associated with sales decisions compared to the single and widowed statuses.

Table 4 presents a descriptive mean comparison of continuous variables between participants and non-participants. All variables that exhibit statistically significant differences reveal higher means for market participant than non-participants. The farmer-household characteristics, education, experience and laborers reveal statistically significant mean differences between the subsamples at $p < 0.01$, while household size is statistically significant at $p < 0.05$. Education and experience enhance farm productivity and marketing through the mastery of skills and knowledge, thus increasing cattle marketing. Experience further captures the effect of social networks and links that accrue over time to enhance the search for trading partners [93]. Thus, the results suggest a positive effect of education and experience on both farmers' sales decisions.

**Table 4.** Participants compared to non-participants based on means of continuous variables.

| Variable | Overall (n = 397) | Participants ($s_1$ = 200) | Non-Participant ($s_2$ = 197) | *t*-Value |
|---|---|---|---|---|
| Herd size | 17.448 (13.220) | 25.145 (14.152) | 9.635 (5.287) | −14.506 *** |
| Steer–heifer ratio | 0.210 (0.176) | 0.226 (0.159) | 0.194 (0.191) | −1.784 |
| Slaughters | 0.514 (0.809) | 0.740 (0.904) | 0.284 (0.623) | −5.858 *** |
| Age | 57.660 (13.178) | 58.835 (12.237) | 56.467 (14.000) | −1.793 |
| Education | 9.388 (4.499) | 10.055 (4.464) | 8.711 (4.443) | −3.007 *** |
| Laborers | 1.408 (0.728) | 1.595 (0.875) | 1.218 (0.472) | −5.353 *** |
| Expenses | 1362.50 (1951.935) | 1992.265 (2405.133) | 723.14 (1004.530) | −6.842 *** |
| Extension | 36.020 (0.173) | 36.030 (0.222) | 36.010 (0.101) | −1.150 |
| Experience | 19.863 (11.774) | 21.935 (10.251) | 17.759 (12.828) | −3.580 *** |
| Household size | 7.942 (3.625) | 8.400 (3.477) | 7.477 (3.721) | −2.554 ** |
| Cows | 7.141 (5.917) | 10.090 (6.808) | 4.147 (2.404) | −11.630 *** |
| Heifers | 2.320 (2.713) | 3.380 (3.122) | 1.244 (1.632) | −8.563 *** |
| Steers | 1.617 (2.208) | 2.455 (2.623) | 0.767 (1.189) | −8.279 *** |
| Pregnant cows | 4.141 (3.861) | 6.045 (4.395) | 2.208 (1.747) | −11.460 *** |
| Calving rate | 0.741 (0.345) | 0.803 (0.250) | 0.678 (0.412) | −3.637 *** |
| Calf mortality rate | 0.214 (0.354) | 0.228 (0.300) | 0.199 (0.402) | −0.790 |

Significance levels: *** $p < 0.01$, ** $p < 0.05$. Standard deviation in parentheses.

Household size and number of laborers are statistically significant at $p < 0.05$ and $p < 0.01$, respectively. Both variables are indicative of the availability of labor required for production and marketing activities. The results suggest that participants have larger households and a greater number of laborers compared to non-participants, thus alluding to the positive effect of these variables on sales decisions.

The statistically significant ($p < 0.01$) cattle-related factors are number of cows, heifers, steers, pregnant cows, slaughters, herd size and calving rate. These factors capture the effects of the herd dynamics that aid productivity, thus promoting the availability of marketable surplus. Concerning

these variables, the larger means for participants suggest a greater propensity for market participation and intensity of market participation compared to non-participants.

Total expenses, with reference to inputs, reflect the capital invested into production and marketing processes. However, marketing expenses reflect observed transaction costs that may inhibit market entry and the extent of participation. The amount of expenses by market participants is statistically and significantly greater than the amount made by non-participants (*Expenses$_{S1}$* = 1992.27, *Expense$_{S2}$* = 723.14). In this case, the expenses are regarded as investment into production processes, since we are dealing with farm-gate sales decisions. Hence, the finding suggests that the variable increases the propensity for cattle marketing.

## 4.2. Econometric Results

The study applied the double-hurdle model to examine the determinants of sales decisions by smallholder beef cattle farmers. The model was estimated using STATA Version 15 and goodness of fit statistics which indicated that the model was significant at $p < 0.01$ with Wald's chi-squared (26) = 397.14, Pseudo R$^2$ = 0.8745 and log pseudolikelihood = −33.824567.

### 4.2.1. Determinants of Market Participation

Table 5 presents the results of the probit regression used for identifying the determinants of market participation. The positive signs attached to the statistically significant parameter estimates indicate a positive effect of each variable on market entry, ceteris paribus. Herd size is significant at $p < 0.01$, reflecting that an increase by one herd of cattle is related to 74.1% proliferation of the probability that a farmer engages in cattle marketing. Previous studies [1,12,30] have found similar findings, but have not discussed market participation in light of sustainability. Since herd size is directly related to productivity and marketable surplus, the implication of the results is that larger herd size increases the probability of market entry through the offtake of readily available market surplus. This in turn improves both the economic and environmental sustainability in cattle production through sustainable use of grazing resources. This finding does not imply the marginalization of smallholder farmers from markets due to their scale of production. In light of population explosion and the dominance of smallholder farmers in the subsector, critical mass production and marketing contribute to the desired increase in food production in sub-Saharan Africa [98]. Moreover, high farm productivity increases marketable surplus, thus cultivating the culture of cattle marketing among smallholder farmers, which minimizes livestock loss of value. This further transforms the culture of wealth preservation into income generation, promoting business suitability of the subsector to ameliorate rural livelihoods. Increased offtake also reduces pasture depletion and land degradation, thereby promoting the stainable use of grazing resources.

Being located in the Lubombo Plateau increases the propensity for market participation by 32.4% ($p < 0.01$). The climatic conditions in the Lubombo Plateau of Eswatini promote the growth of sweetvelds that are highly palatable to beef cattle, bestowing a competitive advantage on farmers in this agro-ecological zone over other climatic zones. High quality grazing contributes to high quality cattle that fetch a high market price, which encourages farmers to engage in active market participation. Moreover, commercial crop production in less in this region, allowing for ample grazing resources for cattle farming. The results are similar to [93] who found market participation to be significantly related to specific farm locations.

Keeping crosses increases the probability of market participation by 11.4% ($p < 0.01$). In the study area, the Brahman breed is perceived to the best exotic breed suitable for crossbreeding with the traditional Nguni cattle. Theoretically, crossbreeding exhibits the advantage of heterosis effect through superior progeny performance, thus increasing productivity and marketable surplus [100]. The implication of the finding is that keeping crossbreds increases productivity and marketable surplus, thus advancing market entry. Market participation increases offtake, thus reducing pasture and land degradation to ensure sustainable beef cattle farming for improved livelihoods. The finding also

presents a pragmatic mechanism of increasing sustainable farm productivity, a rare perspective in current literature on market participation in agriculture.

**Table 5.** Probit regression estimates for determinants of market participation (n = 397).

| Variable | Coef. | Robust Std. Err. | Marginal Effects |
|---|---|---|---|
| Education (Years) | 0.095 | 0.453 | 0.089 |
| Household size (Number) | −0.068 | 0.441 | 0.021 |
| Experience (Years) | 0.788 ** | 0.335 | 0.150 |
| Gender (0 = Female; 1 = Male) | −0.268 | 0.260 | −0.049 |
| Farmer's age (Years) | 0.191 | 1.180 | 0.036 |
| Marital status 1 = Married | 0.259 | 0.448 | 0.049 |
| Marital status 2 = Widowed | 0.009 | 0.537 | 0.002 |
| Off-farm employment (0 = No; 1 = Yes) | 0.747 | 0.605 | 0.139 |
| Ecological zone 1 = Middleveld | −0.212 | 0.230 | −0.040 |
| Ecological zone 2 = Highveld | 0.411 | 0.269 | 0.008 |
| Ecological zone 3 = Lubombo Plateau | 2.002 *** | 0.405 | 0.324 |
| Credit access (0 = No; 1 = Yes) | −0.479 | 0.546 | −0.087 |
| Pasture availability (0 = Insufficient; 1 = Sufficient) | 0.245 | 0.178 | 0.046 |
| Extension (Number) | −35.334 | 28.784 | −6.605 |
| Expenses (Emalangeni) | 0.172 | 0.382 | 0.032 |
| Breed type (0 = Nguni; 1 = Crossbreed) | 0.566 *** | 0.181 | 0.114 |
| Calving rate (Ratio) | 1.568 | 1.012 | 0.293 |
| Slaughters (Number) | 1.182 ** | 0.502 | 0.221 |
| Herd size (Number) | 3.965 *** | 0.651 | 0.741 |
| Steer-Heifer ratio (Ratio) | −0.586 | 1.681 | −0.109 |
| Pregnant cows (Number) | 0.971 * | 0.504 | 0.182 |
| Constant | 47.512 | 44.452 | |

Significance levels: *** $p < 0.01$, ** $p < 0.05$, * $p < 0.1$.

The number of pregnant cows is a "commodity specific feature" that is found to increase the propensity of market participation by 18.2% ($p < 0.1$). Pregnant cows capture the potential for herd size expansion and availability of replacement livestock, thus a motivational factor for market entry since farmers look forward to replacement calves for sold cattle. Selling beef cattle involves the selection of suitable cattle ready for sale considering the dynamics of the herd. Therefore, the implication of this unique finding is that increased cow conception rates and the reproductive superiority of breeding stock increase offtake through market participation. This clarifies the role of market participation in the sustainability discussion and incorporates rural communal cattle farming into the sustainable livestock production framework for the alleviation of hunger and poverty in rural communities.

In Eswatini's smallholder beef cattle farming system, wealth preservation is supreme and home slaughters are rare. The results indicate that an increase by one herd of cattle in home slaughters expands the probability of market entry by 22.1% ($p < 0.05$). Generally, home slaughters are related to distress circumstances such as funerals, otherwise, they are common in blithesome circumstance (weddings, parties, honoring a guest, and so on) with farmers that hold larger herd sizes and have stable sources of income. This is another unique finding of this study, which reveals how herd dynamics influence the marketing behavior of farmers. The results imply that appreciation of the herd's ability to produce marketable surplus promotes a cognitive shift towards increased offtake and market participation, thus transforming communal cattle farming into a sustainably viable enterprise.

The results disclose that a one-year addition on the experience of the farmer is related to a 15.0% ($p < 0.05$) increase in the probability of market participation. A plausible reason for such a result could be that experienced farmers have well-established links and social networks that aid them in soliciting trading partners [93]. Moreover, experience must have taught farmers that holding on to cattle for a long period of time (low market participation) leads to losses through drought, diseases and old age. Another reason could be that experienced farmers are efficient in the production of marketable surplus, thus increased participation in cattle marketing. The result is in line with Egbetokun and Omonona [86] who found that farming experience increases farmers' market participation in food markets.

### 4.2.2. Determinants of the Level of Market Participation

Ceteris paribus, the statistically significant variables allude to an increase or decrease in the level of market participation, subject to the sign of the relevant parameter estimate (see Table 6). The key factors affecting the level of market participation that reveal statistical significance at $p < 0.01$ include extension visits, pasture availability, market channel to individual, married marital status, informal sources of market information, market distance, cows and heifers.

**Table 6.** Truncated regression estimates for determinants of level of market participation ($s_1 = 200$).

| Variable | Coef. | Robust Std. Err. |
|---|---|---|
| Education (Years) | 0.157 ** | 0.061 |
| Household size (Number) | 0.058 | 0.067 |
| Gender (0 = Female; 1 = Male) | −0.020 | 0.030 |
| Experience (Years) | 0.120 ** | 0.049 |
| Marital status 1 = Married | 0.106 *** | 0.038 |
| Marital status 2 = Widowed | 0.085* | 0.049 |
| Non-farm income (0 = <E5000; 1 = ≥E5000) | 0.056 ** | 0.027 |
| Association (0 = No; 1 = Yes) | −0.025 | 0.043 |
| Credit access (0 = No; 1 = Yes) | −0.0356 | 0.027 |
| Calving rate (Ratio) | −0.146 | 0.146 |
| Pasture availability (0 = Insufficient; 1 = Sufficient) | 0.062 *** | 0.023 |
| Extension (Number) | 6.213 *** | 2.018 |
| Breed type (0 = Nguni; 1 = Crossbreed) | 0.032 | 0.023 |
| Cows (Number) | 0.208 *** | 0.056 |
| Heifers (Number) | 0.128 *** | 0.040 |
| Steers (Number) | 0.223 | 0.041 |
| Calf Mortality rate (Ratio) | 0.005 | 0.127 |
| Market distance (Kilometers) | 0.106 *** | 0.032 |
| Market information 1 = Informal sources | 0.279 *** | 0.066 |
| Market channel 1 = Processor | −0.089 * | 0.045 |
| Market channel 2 = Individual | 0.135 *** | 0.046 |
| Market channel 3 = Combination | 0.024 | 0.041 |
| Sale duration 1 = Within a month | −0.011 | 0.029 |
| Expenses (Emalangeni) | 0.099 ** | 0.050 |
| Market price (Average price per herd sold) | −0.197 | 0.135 |
| Laborers (Number) | 0.157 ** | 0.069 |
| Constant | −9.823 *** | 3.196 |

Significance levels: *** $p < 0.01$, ** $p < 0.05$, * $p < 0.1$.

The results suggest that an increase by one extension visit inflates the level of market participation by 6.21%. The finding is in line with some studies that analyzed the intensity of market participation by farmers [41]. Acquiring extra consultations from public and/or private extension and veterinary services reflect a higher level of desire for improved production and marketing. Therefore, the implication of the finding is that seeking more extension and veterinary support improves farm productivity and marketable surplus, thus increasing the extent of market participation. In light of sustainability, intensity of market participation occurs through increased rate of offtake, thus reducing pressure on grazing land to promote environmental sustainability.

Seeking market information through informal sources is related to a 0.28% increase in the intensity of market participation. Since cattle marketing in Eswatini lacks an organized marketing framework, cattle market information through formal channels is scarce. Therefore, establishing intra and inter-communal communication networks enhances exposure to market information, implying an increase in the intensity of market participation. Exposure to market information allows farmers to maximize market incentives by selecting suitable market channels [15]. Then the derived link between market information and sustainability is based on the motivation received by farmers for participating in high incentive-market channels. Such motivation encourages farmers to offer larger volumes of cattle at the market, thereby increasing offtake and intensity of market participation. This then promotes proper use of grazing resources and sustainable rural development through agricultural marketing.

One additional cow increases the quantity of cattle that a farmer is willing to sell by 0.21%. Similarly, an addition of one heifer to the herd is related to a 0.13% increase in the number of cattle offered for sale. Cows and heifers are key herd dynamics that influence the farmers' sale decisions. An increase in the number of cows and heifers captures the availability of replacement stock and ability of the herd to multiply. Therefore, an increase in these dynamics encourages farmers to offer more cattle for sale since there is sufficient back up for breeding stock. Increase in the number of heifers also transforms older and non-productive cows into marketable surplus, thus increasing the extent of market participation. Such empirical results are unique to this study and contribute to existing literature.

Selling cattle to individual persons (market channel 2) is related to a 0.14% increase in the intensity of market participation. The variable reveals a stronger relationship with the sale decision than selling to processors and butchers, which shows statistical significance at $p < 0.1$. Selling to a processor is related to a decline in intensity of market participation by 0.09%. A plausible explanation for these results is based on the principle of bargaining power, where buyers and sellers negotiate the price during farm-gate sales [31]. The nature of distress farm-gate sales erodes farmers' bargaining power when dealing with price-informed purchasing representatives from processors and butcheries, who dominate the cattle market. Farmers often succumb to lower prices offered by such trade partners. Contrary, farmers gain bargaining power when selling to less-price-informed individuals that purchase cattle for purposes such as traditional ceremonies, home slaughters and so on. Such trade partners lack price negotiation skills compared to the experienced representatives of processors and butchers. Therefore, selling to individual buyers increased the intensity of market participation due to better prices.

The results further reveal that being a married farmer is related to a 0.11% ($p < 0.01$) increase in the intensity of market participation. The married status is indicative of a wider human resource base for decision-making, production and marketing functions. Meanwhile, being a widowed farmer is related to a 0.09% increase in the quantity of cattle sold, but at a significance level of $p < 0.1$.

Holding the perception that there is sufficient pasture for grazing is related to a 0.06% increase in the intensity of market participation. Believing that there is enough pastureland and grazing resources motivates farmers to keep larger herd sizes, thereby increasing the intensity of market participation. Logically, the implication is that setting aside sufficient grazing land promotes the farmers extent of market participation. Conversely, when pastures are limited farmers' productivity dwindles and farmers tend to hold on to cattle for longer periods (low intensity of marketing), thus aggravating land degradation. This finding provides a foundation for policy adjustments required by traditional authorities in rural areas for land use allocation that shapes traditional cattle rearing in an environmentally sustainable way.

The positive relationship between the intensity of market participation and market distance implies that an increase by one kilometer in market distance increases the intensity of market participation by 0.11%. Some studies have found a positive effect of market distance on level of market participation [80,94,95]. Noting that free-range cattle farming requires large open grasslands, the further the farmer is from urban areas (large beef processors and butchers), the more grazing land is available. This allows the production of more marketable surplus to warrant high extent of market participation. Another plausible explanation of this counterintuitive result is that buyers may prefer remote farmers who are often out of reach from a variety of buyers. In this case, remoteness promotes the intensification of sales when a buyer is available.

The variables that revealed statistical significance at $p < 0.05$ include education, farming experience, non-farm income, laborers and total expenses. Education enhances farmers' skills and knowledge capacitation, thus improving competence in production and marketing processes. Therefore, an increase by one year of formal education is related to a 0.16% increase in the number of cattle offered for sale, implying that educated farmers sell more cattle than their less educated counterparts. It is expected that educated farmers possess meaningful comprehension of production and marketing processes, thus actively involved in cattle marketing [83].

One additional person in the number of laborers relates to a 0.16% inflation in the quantity of cattle sold. The results are consistent with Kefyalew [91] who found a strong positive relationship between extent of market participation and family labor. In the study area, it is a traditional responsibility of young males (family labor) to look after beef cattle, and households with more labor have the wide ability to manage large herds, thus high marketable surplus and extent of market participation.

The results further reveal that an additional year of a farmer's experience increases the intensity of market participation by 0.12%. Experience bestows production skills and knowledge that are useful in production and marketing processes [91]. As discussed before, experience provides critical lessons against low market participation, thus increased extent of market participation. Moreover, experienced farmers are efficient in the production of marketable surplus, thus increased level of market participation.

Moreover, earning non-farm income greater or equal to E5000 relates to a 0.06% inflation in the quantity of cattle sold. Contrary to Musah et al. [11], who found a negative impact of non-farm income, the positive results of this study infer that beef cattle farmers invest off-farm income as capital into production and marketing functions, thus increasing the level of market participation. Similarly, the addition of one Emalangeni on capital used to cover expenses is related to a 0.10% increase in the intensity of market participation.

## 5. Conclusions and Policy Implications

### 5.1. Conclusions

The descriptive assessment of the study reveals significant differences between market participants and non-participants. Most importantly, participants keep crossbred cattle, which allow them to increase herd sizes, the number of cows as well as pregnant cows, thereby increasing the calving rate. This further induces significant difference in the number of heifers and steers, providing a more marketable surplus compared to non-participants. Significantly more participants were also found to have access to farm credit, and had more non-farm income than non-participants did. This induced a significant difference with respect to the amount of money used to solicit inputs, thus improving productivity and market participation. Participants were also found to be significantly more educated than their counterparts, promoting their mastery of skills and knowledge in production and marketing processes. Significant differences were also revealed in farming experience, number of laborers, household size, marital status and gender. As expected, male farmers participate in more cattle marketing activities than females.

The econometric analysis indicated that the increase in the probability of market participation is mainly related to increases in production shifter and cattle-related factors. Keeping crossbreeds revealed a significant impact on market participation. Crossbreeds have a heterosis advantage that increases productivity, thus increasing herd size and conception rates. These in turn increase marketable surplus and market participation. Further significant impacts were found in relation to farming experience and ecological zone.

The level of market participation significantly relates to the quantity of cows and heifers in the herd. These indicate potential for replacement stock, thus increased marketable surplus and intensity of market participation. Pasture availability and extension services revealed a significant positive effect on the number of cattle offered for sale. The significant effect of non-farm income may have also induced the positive significant impact from the total amount of money used for soliciting inputs and services. The results further revealed that farmers have an improved bargaining power when selling to individual persons than when dealing with processors and butchers.

The farmer and household characteristics that indicated significant effects on the number of cattle offered for sale were farming experience, marital status, number of laborers and education. These variables increase the level of market participation, holding all other factors constant. The

market factors, market distance and access to market information, were also found to impose a positive impact on the intensity of market participation.

*5.2. Policy Implications*

The synthesis of the results discloses that increased cattle marketing mainly depends on farmers' capacity to produce marketable surplus. First, farmers should be encouraged to crossbreed the small-sized Nguni cattle with the fast-growing and large-sized exotic breeds. This is necessary to improve the breeding performance and quality of cattle, thus increasing marketable surplus. Second, decentralization of veterinary services should be implemented to ease the cost of acquiring more consultations necessary for improved production. Moreover, extension service adjustments should be implemented to provide training on commercial cattle farming, pricing and marketing of cattle. This can enrich farmers' skills and knowledge critical for efficient use of production resources to increase marketable surplus. The extension services department can also function as a communication channel for price and market information to assist farmers in the selection of market channels that maximize market incentives.

Rural and national development frameworks should enhance the establishment of market structures and policies that advance institutional support and market flow within the subsector. Institutional frameworks should aim at establishing integrative linkages between smallholder farmers and commercial enterprises to promote scale economies and vertical integration in cattle marketing. Such development frameworks should be diverse, incorporating capital mobilization strategies, such as cooperatives and group ranching, to enhance farmers' access to input resources that are necessary to increase the production of marketable surplus. Finally, development programs should embrace female empowerment as means of encouraging female farmers to participate and successfully compete with their male counterparts in cattle marketing. Addressing these recommendations would not only promote the business sustainability aspect of the subsector, but also enhance the transition towards an environmentally sustainable cattle production system by reducing pasture and land degradation induced by low market participation.

**Author Contributions:** Conceptualization, methodology and validation, S.I.D. and W.-C.H.; formal analysis, investigation, data curation and writing—original draft preparation, S.I.D.; writing—review and editing, S.I.D. and W.-C.H.; supervision, W.-C.H.

**Funding:** This research received no external funding.

**Conflicts of Interest:** The authors declare no conflicts of interest.

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
