# Peer review of "A Double Hurdle Estimation of Sales Decisions by Smallholder Beef Cattle Farmers in Eswatini"

_sustainability, doi:10.3390/su11195185_

Round 1

Reviewer 1 Report

- First of all I think Authors should clarify how the paper fits with journal's aim and scope. There is just a small phrase where they recall a link, but I think they must provide a more detailed explanation. 

- Introduction has to be rewritten, by introducing key elements, like: aim and scope of the paper, research questions, originality of the paper, that is authors’ contribution to literature. Authors should specify how this topic has been treated in literature and how authors’ would like to put forwards their analysis with respect to literature. The description of the cattle sector cannot be treated in the introduction.

- Please explain what smallholder means according to authors (is there a classification according to farm’s size?). Which classification authors make reference to? The European one or other classification?

- Lines 123-124: the 4 districts considered by authors are homogeneous under a territorial point of view? A brief description of rural areas under study could be of help, provided that in the results there is one district where the results are emphasised with respect to others. 

- Lines 153-154: authors should be more precise in explaining how they define transaction costs and how they intend to take them into consideration. Institutional factors is a complex variable that require more specification. Moreover, do they consider institutional factors just limited to the role of extension services? If so, which kind of services, public or private? This is of paramount importance, in that number of visits could imply a cost for smallholders if provided by private agents.

- the “I” in formula (2) needs for deeper clarification.

Please provide more details to understand links between formula 2 and variables used in table 2.

- Distance to market is a relevant variables. Authors should provide more information about type of market, what is market? What is the dimension of the market in terms of potential consumers? Are they dealing with an urban centre, a big or a small town? In the results paragraph, lines 344-345 authors make reference to farm location, this should be specified also in methodological paragraph.

- Results

Some results seem not surprising (level of education, farm size, etc.), authors should underline similarities and dissimilarities with other analyses.

Lines 330-332: The implication is that holding bigger herd size increases 330 the propensity of participating in cattle markets [51]. Ideally, bigger herd size is associated with the 331 availability of marketable surplus, hence increased probability of market entry. Of course it is, this is the basic concept of critical mass of marketing.

Therefore, results seem poor nothing new is found by author; if so, authors should emphasise the novelty of their paper.

- Policy implication: when authors say extensionists provide support for market participation, do they consider that jointly with the relevance of farm size in accessing market participation may discriminate smaller farms?

Smaller farms are excluded from markets or are there any opportunities for them to enter the market? Authors should mention eventual collective action to be realized for entering the market in a more robust way, like producers organizations or cooperatives. 

- Other 

Figure 1: “populaition”

Figure 1: the title is repeated below and inside the figure

Reviewer 2 Report

A Double Hurdle Estimation  of Sales Decisions by Smallholder Beef Cattle Farmers in Eswatini.

This is an interesting paper which tries to explain the estimation of sales decisions by smallholder beef cattle farmers, in Eswatini, (Swazilandia, in South Africa).  

Although the paper certainly presents some interests, some theoretical corrections need to be made such as:

Econometric foundations of the study needs to be improved. The data description variables needs to also to be improved. Descriptive variables such as mean, mode and standard deviation should have to be introduced. A data sources description should have to be introduced. The theoretical description of the farmer-house socio-economic cattle related factors, needs to be clearly improved (a definition of the descriptive variable is needed). A definition of variables which are included in the level of market participation model needs to be intoduced. A theoretical desciption to explain the use of the reduced form of the market participation model is needed. The theoretical discussion about the use of probit model and the results needs to be deeple improved. The theoretical foundations of the funcions which are used to explain the theoretical models needs clearly to be improved (specially in the description of the related literature).  

Reviewer 3 Report

The manuscript entitled “A Double Hurdle Estimation of Sales Decisions by Smallholder Beef Cattle Farmers in Eswatini” is dedicated to analyze the determinants of farmers’ sales decisions in cattle marketing. Authors carried out a survey with a semi-structured interview dedicated to farmers (sample n=397) and applied descriptive statistics and double hurdle model on cross-sectional survey’s data.

In my humble opinion, the aim of this paper is interesting and worthy to investigate. The introduction is quite clear and describes the subject treated in a good manner. The analysis of the literature is congruous and explains the subject of the paper and related factors. Findings provide useful information to the scope of the paper allowing an appropriate and consistent presentation of the main issues in the last paragraph. On the whole, I consider this paper well-written, however, I think that some suggestions could be useful to improve the quality of the paper i.e.

more information on Eswatini should be provided to improve the knowledge of the study area e.g. the authors could be carried out a subparagraph dedicated to territorial framework; the description of the questionnaire should be expanded e.g. integrating the provided information (lines 130-135) and/or inserting the questionnaire as an annex; authors should explain more in-depth the phase of farmers’ interviews and provide references to justify the selected method; authors cited references “52-56” in the discussion paragraph but are not presented and examined in the introduction paragraph. In my opinion, these references should be describe in the introduction paragraph and then discuss in comparison with the findings of the study. Therefore, I suggest to emphasize the relevance of the references 52-56 implementing their descriptions in the introduction paragraph e.g. carrying out a sub-paragraph as Literature review; the references should be integrated with more recent sources; Lastly, some misprints occur e.g. lines 18 “famers”, Figure 1 “population”, captions Table 3 and Table 4 “ . ”.

On the basis of aforementioned comments, the present form of this paper should be revised to be eligible for the publication on IF journal as Sustainability. I think that authors should integrate the contents of the paper with detailed information and partially restructure the body of the paper.

Round 2

Reviewer 1 Report

The paper has been deeply modified, according to my suggestions. As far as my knowledge is concerned, the reviewed version of the paper is suitable for publication.

Best regards

Reviewer 2 Report

 -